# Frequency of multiple changes to prespecified primary outcomes of clinical trials completed between 2009 and 2017 in German university medical centers: A meta-research study

Martin Holst[1,2‡]*, Martin Haslberger[1‡], Samruddhi Yerunkar[1], Daniel Strech[1], Lars G. Hemkens[3,4,5,6‡], Benjamin G. Carlisle[1‡]

1 QUEST Center for Responsible Research, Berlin Institute of Health at Charité–Universitätsmedizin Berlin, Berlin, Germany, 2 Institute for Ethics, History and Philosophy of Medicine, Medizinische Hochschule Hannover, Hannover, Germany, 3 Department of Clinical Research, University Hospital Basel, University of Basel, Basel, Switzerland, 4 Meta-Research Innovation Center Berlin, QUEST Center for Responsible Research, Berlin Institute of Health at Charité–Universitätsmedizin Berlin, Berlin, Germany, 5 Meta-Research Innovation Center at Stanford, Stanford University, Stanford, California, United States of America, 6 Pragmatic Evidence Lab, Research Center for Clinical Neuroimmunology and Neuroscience (RC2NB), University Hospital Basel and University of Basel, Basel, Switzerland

‡ MH and MH share first authorship on this work. LGH and BGC are joint senior authors on this work.
* martin.holst@bih-charite.de

## Abstract

### Background

Clinical trial registries allow assessment of deviations of published trials from their protocol, which may indicate a considerable risk of bias. However, since entries in many registries can be updated at any time, deviations may go unnoticed. We aimed to assess the frequency of changes to primary outcomes in different historical versions of registry entries, and how often they would go unnoticed if only deviations between published trial reports and the most recent registry entry are assessed.

### Methods and findings

We analyzed the complete history of changes of registry entries in all 1746 randomized controlled trials completed at German university medical centers between 2009 and 2017, with published results up to 2022, that were registered in ClinicalTrials.gov or the German WHO primary registry (German Clinical Trials Register; DRKS). Data were retrieved on 24 January 2022. We assessed deviations between registry entries and publications in a random subsample of 292 trials. We determined changes of primary outcomes (1) between different versions of registry entries at key trial milestones, (2) between the latest registry entry version and the results publication, and (3) changes that occurred after trial start with no change between latest registry entry version and publication (so that assessing the full history of changes is required for detection of changes). We categorized changes as major if primary outcomes were added, dropped, changed to secondary outcomes, or secondary outcomes were turned into primary outcomes. We also assessed (4) the proportion of publications

**Data Availability Statement:** The final dataset has been posted to our Open Science Framework page (https://osf.io/e2uct/). Analysis code and original datasets are available on Github (https://github.com/Martin-R-H/InvisibleOutcomeChanges).

**Funding:** This work was funded under a grant from the Federal Ministry of Education and Research of Germany (Bundesministerium fuer Bildung und Forschung – BMBF) [01PW18012], awarded to DS. The funder had no role in study design, data collection and analysis, decision to publish, or preparation of the manuscript.

**Competing interests:** The authors have declared that no competing interests exist.

transparently reporting changes and (5) characteristics associated with changes. Of all 1746 trials, 23% (n = 393) had a primary outcome change between trial start and latest registry entry version, with 8% (n = 142) being major changes, that is, primary outcomes were added, dropped, changed to secondary outcomes, or secondary outcomes were turned into primary outcomes. Primary outcomes in publications were different from the latest registry entry version in 41% of trials (120 of the 292 sampled trials; 95% confidence interval (CI) [35%, 47%]), with major changes in 18% (54 of 292; 95% CI [14%, 23%]). Overall, 55% of trials (161 of 292; 95% CI [49%, 61%]) had primary outcome changes at any timepoint over the course of a trial, with 23% of trials (67 of 292; 95% CI [18%, 28%]) having major changes. Changes only within registry records, with no apparent discrepancy between latest registry entry version and publication, were observed in 14% of trials (41 of 292; 95% CI [10%, 19%]), with 4% (13 of 292; 95% CI [2%, 7%]) being major changes. One percent of trials with a change reported this in their publication (2 of 161 trials; 95% CI [0%, 4%]). An exploratory logistic regression analysis indicated that trials were less likely to have a discrepant registry entry if they were registered more recently (odds ratio (OR) 0.74; 95% CI [0.69, 0.80]; p<0.001), were not registered on ClinicalTrials.gov (OR 0.41; 95% CI [0.23, 0.70]; p = 0.002), or were not industry-sponsored (OR 0.29; 95% CI [0.21, 0.41]; p<0.001). Key limitations include some degree of subjectivity in the categorization of outcome changes and inclusion of a single geographic region.

## Conclusions

In this study, we observed that changes to primary outcomes occur in 55% of trials, with 23% trials having major changes. They are rarely transparently reported in the results publication and often not visible in the latest registry entry version. More transparency is needed, supported by deeper analysis of registry entries to make these changes more easily recognizable.

**Protocol registration**: Open Science Framework (https://osf.io/t3qva; amendment in https://osf.io/qtd2b).

---

## Author summary

### Why was this study done?

- Clinical trial registries are a key tool to increase the trustworthiness of clinical trials. They allow assessment of how closely a published trial follows its original plan.

- However, registry entries can be updated at any time, which creates a trail of historical versions. If the latest registry entry version matches with the published trial report, important preceding changes might thus be unapparent to assessors at first glance.

- Our objective was to investigate how often primary outcomes are changed in the trial registry over the course of a trial, and how often outcome changes are unapparent if one compares only the latest registry entry version to the publication.

**What did the researchers do and find?**

- We assessed all 1746 randomized controlled trials completed at German university medical centers between 2009 and 2017, that have a results publication and that had been registered in either an international or a German clinical trial registry. We determined the frequency of outcome changes between different versions of a registry entry, as well as the latest registry entry and the results publication.

- We defined adding or dropping primary outcomes, changing them to secondary outcomes, or turning secondary outcomes into primary outcomes, as major changes.

- We found that approximately 55% of trials had primary outcome changes at any timepoint over the course of a trial; 23% of trials had major changes. We observed changes that can be easily identified by comparing the published results to the latest registry entry in 41% of trials. In 14% of trials, however, the changes would require an in-depth look within the historical versions of that trial's registry entry.

- Only 1% of trials with changes (2 trials) reported this in the corresponding publications.

**What do these findings mean?**

- Our analysis suggests that changes to primary outcomes of a clinical trial are common, are often major, and have a potential to go unnoticed.

- More transparency is needed, supported by deeper analysis of registry entries to reveal these outcome changes.

## Introduction

Clinical trial registries are tools to enhance the transparency and trustworthiness of clinical trial evidence by openly reporting key study parameters in a rapidly accessible manner [1]. They make it possible to detect selective reporting biases that pose a threat to validity, by allowing reviewers to assess possible deviations from an original protocol and analysis plan. One of the most critical deviations is a change to prespecified outcome measures (or 'outcomes', for short). Consequently, outcomes and their pre-specification are among the most important reporting items of clinical trials, with the CONSORT statement asking authors to describe 'completely defined pre-specified primary and secondary outcome measures, including how and when they were assessed' as well as 'any changes to trial outcomes after the trial commenced, with reasons' [2]. If readers do not know whether there have been any changes to the outcomes, e.g., by adding or dropping primary outcomes, they cannot assess the risk of bias due to selective reporting of only statistically significant results or multiple hypothesis testing [3].

Many clinical trial registries, such as ClinicalTrials.gov, allow entries to be updated after initial registration. Updating a registry entry is a useful tool, as parts of a trial protocol often change and it is desirable for an entry to reflect the most current information. It still constitutes another possible source of reporting bias if entries are not properly maintained and larger changes go unreported. While a number of ethical and reporting guidance documents

recommend prospective trial registration [2, 4, 5] and some recommend the transparent reporting of protocol changes [2, 5], this is only checked in a minority of cases by peer-reviewers [6]. But even if trial registries were considered during review, looking only at the latest registry entry version may lead to an incomplete assessment, as changes over the course of the trial would remain unapparent if the latest registry entry matches with the results publication. These changes would only be discovered by closely investigating the history of changes in trial registries, which some trial registry platforms maintain. These histories, however, are often not easily accessible.

We systematically searched studies that have assessed the frequency of outcome changes between registries and publications [7]. We identified several analyses which widely agree that this practice is common, with a median estimated prevalence of 31% (interquartile range (IQR): 17–45%) of clinical trials affected [8]. However, most studies did not report which registry entry version they used [9–16], or reported that they only assessed changes between the publication and one single registry entry version (i.e., the latest available entry [17–25], the latest entry before trial completion [26], the first available entry [27], or the first entry during the active phase [28]). We are aware of only two studies that examined changes to primary outcomes over the course of the entire registry version history. Both used relatively small and selective samples (either trials published in ICMJE journals or trials registered on ClinicalTrials.gov covered by the Food and Drug Administration Amendments Act) over a four-year time period [29, 30]. We are not aware of studies quantifying the frequency of changes that may go potentially unnoticed because the latest registry entry and the results publication match.

In this study, we aimed to provide an in-depth analysis of changes to primary outcomes using a comprehensive nationwide sample over 9 years that represents the current practice of trial registration in academic research. Specifically, we aimed to assess how often primary outcomes of clinical trials were changed (1) across all clinical trial registry entry versions and (2) between the latest registry entry and the results publication; how often (3) changes are not detectable when only the latest registry entry version would be inspected, and (4) how often they are transparently reported in the results publications; and (5) which trial characteristics are associated with these outcome changes.

## Methods

### Data sources and sample

We based our analyses on previous work of our group that provided the highly granular and large-scale data on both registry entries and accompanying results publications required to address our aims. Two large datasets [31, 32] of randomized controlled trials covering a period of 9 years of trial completion plus a period of 5 years of publication tracking were used for this project. The trials were completed at German university medical centers between 2009 and 2017 and had been registered in either ClinicalTrials.gov or the German World Health Organization primary registry (German Clinical Trials Register or *Deutsches Register Klinischer Studien*; DRKS). Both registries have an accessible history of changes. The corresponding results publications were identified in manual searches up to 2022, ensuring an at least five-year period after trial completion and complementing the often inconsistent references in the trial registry [33]. We retrieved a combined dataset for both projects from a GitHub repository [34] on 24 January 2022.

### Eligibility criteria

We included any study that fulfilled all of the following criteria: 1) has a registry entry in either the ClinicalTrials.gov or the DRKS database, 2) was completed between 2009 and 2017

according to the registry trial status described as 'Completed', 'Unknown status', 'Terminated', or 'Suspended' (ClinicalTrials.gov), or 'Recruiting complete, follow-up complete', 'Recruiting stopped after recruiting started', or 'Recruiting suspended on temporary hold' (DRKS), 3) reported in the registry that a German university medical center was involved (i.e., mentioned as responsible party, lead/primary sponsor, principal investigator, study chair, study director, facility, collaborator, or recruitment location; for definitions see [31, 32]), 4) has published results, i.e., a full-text publication of the trial results was found by the search methods described in the protocol for the underlying datasets [31, 32], 5) is registered as a randomized trial.

## Data extraction and processing

We used a stepwise approach of automatic data extraction and processing followed by reviewer assessment of identified outcome changes. We first compared outcomes between different registry entry versions, followed by comparing the most recent registry entries to the accompanying results publications.

**Data extraction and review procedure.** Trial identifiers (NCT or DRKS number) and basic information about the trials and their accompanying results publications were included in the dataset [31, 32]. We then retrieved all historical registry entry versions of these trials, using R 4.1.3 [35] and the cthist R package [36], on 24 January 2022 (DRKS) and 12 April 2022 (ClinicalTrials.gov). We extracted data from the fields listed in the codebook [37]. For our classification of within-registry changes, we extracted primary and secondary outcomes at four key trial milestones: (1) the first entry after study start (start date; inclusion of the first patient), (2) the latest entry before the end of active status (completion date; determined by 'completed' status in the registry), (3) the latest entry before results publication (publication date), and (4) latest available entry.

We classified trials into medical fields based on their journal's category from the SCImago Journal & Country Rank database [38], which is based on the Scopus medical field classification. The journals' categories were divided into 17 higher-order categories based on consensus (MH, MH, LGH).

**Assessment of within-registry changes.** Two reviewers (MH, MH) assessed within-registry changes in duplicate and resolved conflicts through discussion, using the Numbat Systematic Review Manager [39]. Where necessary, a third reviewer (LGH) resolved conflicts. Reviewers assessed changes to the primary outcomes between the four key trial milestones within the registry (Fig 1), using a classification system that categorizes changes to primary outcomes as either major changes (based on an existing classification system [40]) or minor changes (adapted from [20, 41]). Major changes were defined as adding or dropping primary outcomes, changing them to secondary outcomes, or turning secondary outcomes into primary outcomes. Minor changes were defined as changes to the outcome's type of measurement, metric or method of aggregation, or timing of assessment, or as either added specificity or omissions to these (for more detail see Table 1).

**Assessment of registry-publication changes.** For feasibility, we drew a random sample of 300 trials with accompanying results publications, from which three reviewers (MH, MH, SY) extracted primary outcomes. Eight articles were excluded from the sample because they were not the main results publications (i.e., the final sample size was 292). We considered outcomes primary if they were explicitly named as such using a list of keywords ('primary', 'main', 'outcome', 'endpoint', 'end point'). Otherwise, we used the outcome used for sample size calculation, if reported. In all other cases, we used the first reported outcome in the abstract and results section (with priority given to the abstract). For each entry, reviewers compared the

**Registry Entry Versions**

**Fig 1. Overview of timepoints of outcome changes assessed in trial registries and corresponding results publications.** Registry entry versions at four subsequent key trial timepoints were assessed (blue box). Changes can occur between registry entry versions (within-registry changes; red arrows) and between the latest registry entry and results publication (blue arrow). In the lower part, examples of primary outcome descriptions are given (pink boxes), with corresponding categorizations of changes in grey next to the arrows (see Table 1 for more details on categorization).

extracted primary outcome(s) to the latest available registry entry version using the developed rating system (Fig 1 and Table 1). Reviewers also recorded whether there were any mentions of outcome changes in the assessed results publications. Conflicts were resolved through discussion between reviewers, with one additional reviewer (LGH) resolving conflicts, where necessary.

**Inter-rater agreement.** The overall inter-rater agreement before discussion, over all timepoints and items (measured by Cohen's kappa), was $\kappa = 0.81$ for the within-registry change ratings and $\kappa = 0.46$ for the registry-publication change ratings. Disagreement resulted partly from cases without clear identification of the primary outcomes in results publications (if only the explicitly mentioned outcomes were used for the registry-publication change ratings, inter-rater agreement was $\kappa = 0.51$). In a post-hoc sensitivity analysis we restricted our analysis to primary outcomes that were explicitly named as such in the results publications or had been used for sample size calculation.

## Statistical analyses and reporting

We report descriptive statistics for the frequency of different types of within-registry outcome changes (between key trial milestones) and registry-publication outcome changes. For percentages within the random sample of 292 trials with manual assessment of results publications, we report 95% confidence intervals (CI). For the analysis of determinants of within-registry outcome changes or registry-publication outcome changes, we used logistic regressions, with either any within-registry outcome change, or any registry-publication outcome change, as output variables, and a set of 9 candidate input variables (study phase, sponsor,

**Table 1. Categories of primary outcome changes*.**

| Major changes | |
|---|---|
| ▪ New primary outcome introduced | A new primary outcome was introduced. This category also applied when (seemingly) composite outcomes or different timepoints of measurement were split into separate primary outcomes. |
| ▪ Primary from secondary | A new primary outcome was introduced but had been listed as a secondary outcome before. |
| ▪ Primary outcome omitted | A previously registered primary outcome was completely omitted. |
| ▪ Primary to secondary | A previously registered primary outcome was reported as a secondary outcome in the later registry entry or publication. This also applied when (seemingly) composite outcomes were split into separate outcomes and some were later listed as secondary outcomes. |
| **Minor changes** | |
| ▪ Type of measurement changed, ▪ metric or method of aggregation changed, ▪ timing of assessment changed | Details of a primary outcome's type of measurement, metric or method of aggregation or timing of assessment were changed, e g., follow-up timing changed from "24h" to "48h". |
| ▪ Type of measurement specified, ▪ metric or method of aggregation specified, ▪ timing of assessment specified | Details of a primary outcome's type of measurement, metric or method of aggregation or timing of assessment were specified for the first time or had a significant detail added, e.g., from "seizure rate" to "seizure rate as recorded by family members". |
| ▪ Type of measurement omitted, ▪ metric or method of aggregation omitted, ▪ timing of assessment omitted | Details of a primary outcome's type of measurement, metric or method of aggregation or timing of assessment were completely omitted or described with less specificity, e.g., from "seizure rate as recorded by family members" to "seizure rate". |
| **No changes** | |
| ▪ No relevant change | We did not consider, for example: <br>• Correction of typos, <br>• description of statistical analyses (e.g., intention-to-treat vs per-protocol population, statistical tests used), <br>• description of handling of missing data, <br>• addition of redundant descriptions of commonly known scales in the field (e.g., Response Evaluation Criteria in Solid Tumors, Visual Analogue Scale), or <br>• redundant details pertaining to measurement (e.g., change from baseline). |

* These categories were used to classify changes between different versions of a registry entry, or between the latest registry entry and the results publication (see Fig 1).

publication year, registration year, medical field, registry, multicenter trial, enrollment, intervention). These variables were prespecified before the start of regression analyses (details in S1 and S2 Tables). We used the PRISMA statement [42] to structure our manuscript (S1 PRISMA Checklist).

## Results

We included 1746 clinical trials completed at German university medical centers between 2009 and 2017 (Table 2 and Fig 2). The median sample size was 100 (IQR: 50–264), their median registration year was 2011 (IQR: 2009–2013) and median publication year was 2015 (IQR: 2013–2017). Most trials investigated topics in general medicine (34%) and internal medicine (19%). Industry was the lead sponsor for 26% of trials.

### Prevalence of outcome changes

**Primary outcome changes within trial registry entries.** Of 1746 trials, 393 (23%) had a primary outcome change between trial start and latest registry entry version, i.e., within registry entry versions between key trial milestones (Table 3 and S1 Fig). Minor changes were

**Table 2. Characteristics of all 1746 randomized controlled trials completed between 2009 and 2017 in German university medical centers, and the random sample of 292 randomized controlled trials.**

| Characteristic | | Population (N = 1746) | Sample (n = 292) |
|---|---|---|---|
| | | Median (IQR) | Median (IQR) |
| **Registration Year** | | 2011 (2009, 2013) | 2011 (2009, 2013) |
| **Publication Year** | | 2015 (2013, 2017) | 2015 (2013, 2017) |
| **Sample Size** | | 100 (50, 264) | 100 (48, 269) |
| | | Number of trials (%) | Number of trials (%) |
| **Medical Field** | General Medicine | 598 (34%) | 93 (32%) |
| | Internal Medicine | 333 (19%) | 65 (22%) |
| | Neuroscience | 121 (7%) | 19 (7%) |
| | Pharmacology, Toxicology and Pharmaceutics | 115 (7%) | 20 (7%) |
| | Surgery | 103 (6%) | 23 (8%) |
| | Oncology | 97 (6%) | 16 (5%) |
| | Psychology and Psychiatry | 83 (5%) | 13 (4%) |
| | Family & Reproductive Medicine | 50 (3%) | 12 (4%) |
| | Dentistry | 25 (1%) | 3 (1%) |
| | Nursing | 19 (1%) | 1 (0%) |
| | Health Professions | 16 (1%) | 4 (1%) |
| | Immunology and Microbiology | 15 (1%) | 1 (0%) |
| | Epidemiology and Public Health | 13 (1%) | 4 (1%) |
| | Basic | 10 (1%) | 2 (1%) |
| | Other Clinical Field | 41 (2%) | 3 (1%) |
| | Other Medical Field | 33 (2%) | 4 (1%) |
| | Other | 74 (4%) | 9 (3%) |
| **Study Phase** | None/not applicable | 862 (49%) | 154 (53%) |
| | Phase: 1 | 56 (3%) | 6 (2%) |
| | Phase: 2 | 268 (15%) | 38 (13%) |
| | Phase: 3 | 382 (22%) | 62 (21%) |
| | Phase: 4 | 178 (10%) | 32 (11%) |
| **Sponsor** | Industry | 455 (26%) | 69 (24%) |
| | Other | 1291 (74%) | 223 (76%) |
| **Registry** | ClinicalTrials.gov | 1402 (80%) | 240 (82%) |
| | DRKS | 344 (20%) | 52 (18%) |
| **Multicenter Trial** | No | 930 (53%) | 155 (53%) |
| | Yes | 815 (47%) | 137 (47%) |
| **Intervention** | Device | 281 (16%) | 47 (16%) |
| | Drug or Biological | 631 (36%) | 96 (33%) |
| | Other or not provided | 834 (48%) | 149 (51%) |

found in 318 (18%) trials. 142 trials (8%) had major changes, i.e., primary outcomes were added, dropped, changed to secondary outcomes, or secondary outcomes were turned into primary outcomes. Of these major changes, 66 (4%) happened between first patient inclusion (start date) and trial completion date, 49 (3%) between completion and publication date, and 36 (2%) between the publication date and the latest registry entry.

**Primary outcome changes between latest registry entry and publication.** Primary outcomes in publications were different from the latest registry entry version in 41% of trials (120 of the 292 sampled trials; 95% CI [35%, 47%]), with major changes in 18% (54 of 292; 95% CI [14%, 23%]) and minor changes in 26% (75 of 292; 95% CI [21%, 31%]) (Table 3).

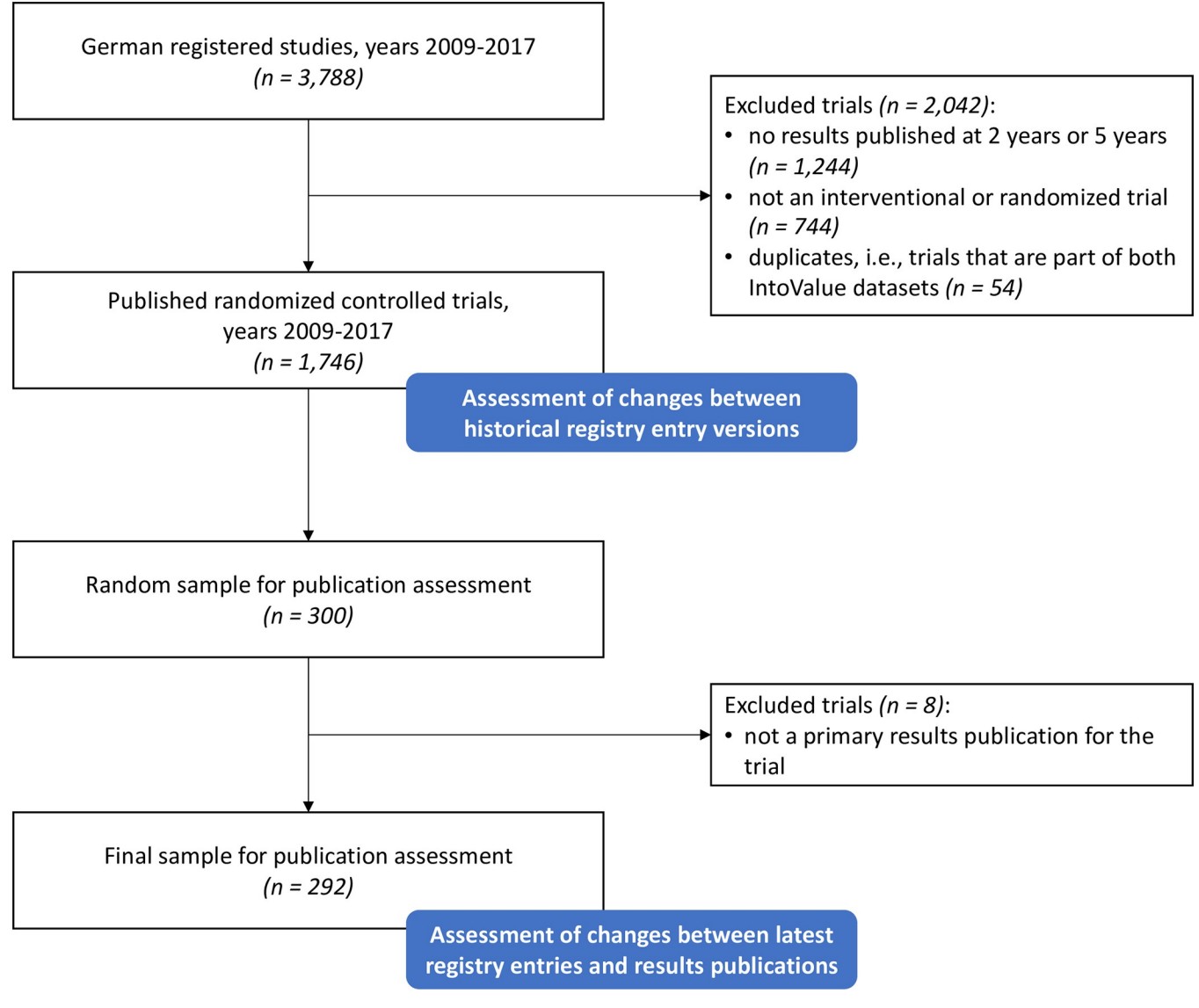

**Fig 2. Study flowchart.**

### Apparent and unapparent changes of primary outcomes

Among the 292 trials, 41 trials had changes only within the registry histories while the latest registry entry matched the results publication (14%; 95% CI [10%, 19%]). Of these changes, 13 were major (4%; 95% CI [2%, 7%]) and 35 minor (12%; 95% CI [8%, 16%]). Changes between the latest registry entry and the results publication, but also within the registry were found in 25 trials (9%; 95% CI [6%, 12%]), of which 12 (4%; 95% CI [2%, 7%]) had major and 14 (5%; 95% CI [2%, 7%]) had minor changes. Overall, 161 trials had any change to their primary outcomes after study start (55%; 95% CI [49%, 61%]), with 67 having major changes (23%; 95% CI [18%, 28%]) and 110 having minor changes (38%; 95% CI [32%, 44%]) (Figs 3, S2, and S3).

### Reporting of changes in the publications

Of all 161 trials with any change of a primary outcome (i.e., 120 trials with easily detectable changes between the latest registry entry and the publication and 41 trials with more

**Table 3. Primary outcome changes in 1746 randomized controlled trials completed between 2009 and 2017 in German university medical centers.**

| | Changes within registry entries (n = 1746) Number of trials (%) | | | | Registry-publication changes[7] (n = 292) Number of trials (%) [95% CI] |
|---|---|---|---|---|---|
| | Any[2] | Start date[3] vs Completion date[4] | Completion date[4] vs Publication date[5] | Publication date[5] vs Latest entry[6] | |
| **Changes[1]** | | | | | |
| **any** | **393 (23%)** | **167 (10%)** | **159 (9%)** | **131 (8%)** | **120 (41%) [35%, 47%]** |
| major | 142 (8%) | 66 (4%) | 49 (3%) | 36 (2%) | 54 (18%) [14%, 23%] |
| • addition of primary outcomes | 121 (7%) | 54 (3%) | 42 (2%) | 31 (2%) | 36 (12%) [9%, 17%] |
| • deletion of primary outcomes | 57 (3%) | 25 (1%) | 19 (1%) | 14 (1%) | 34 (12%) [8%, 16%] |
| minor | 318 (18%) | 117 (7%) | 130 (7%) | 110 (6%) | 75 (26%) [21%, 31%] |
| • changes to relevant details | 149 (9%) | 49 (3%) | 61 (3%) | 51 (3%) | 45 (15%) [11%, 20%] |
| • addition/omission of relevant details | 233 (13%) | 78 (4%) | 91 (5%) | 80 (5%) | 32 (11%) [8%, 15%] |
| **no information in registry[8]** | **5 (0%)** | **332 (19%)** | **266 (15%)** | **945 (54%)** | **0 (0%) [0%, 0%]** |
| **none** | **1348 (77%)** | **1247 (71%)** | **1321 (76%)** | **670 (38%)** | **172 (59%) [53%, 65%]** |

CI: Confidence interval.

1: Multiple changes may occur at different timepoints. A trial can have both major and minor changes. Categories in boldface are mutually exclusive and add up to 1746 or 292, respectively.

2: Any changes to primary outcomes reported in the registries at different trial timepoints (within-registry changes).

3: Start date (i.e., the registry entry version at the time of first patient inclusion).

4: Completion date (i.e., the last registry entry version before the primary completion date).

5: Publication date (i.e., the registry entry version at the date of the first results publication).

6: Latest entry (i.e., the most recent registry entry version by the time we retrieved the data).

7: Registry-publication changes refer to differences between the latest registry entry and the published paper.

8: Milestone does not exist in registry.

unapparent changes before), 2 trials reported this change in the results publication (1%, 95% CI [0%, 4%]). One of the trials, which exhibited a major change both within the registry (between completion and publication date) and between the latest registry entry version and the results publication, stated that some of the primary outcomes were going to be reported in a separate manuscript [43]. The other publication, which had a major change between the latest registry entry version and the results publication, stated that the changed outcome was the 'more sensitive and direct measurement' [44].

## Trial characteristics associated with outcome changes (within-registry and registry-publication)

Within-registry outcome changes were less likely in non-industry-sponsored trials (14% Non-Industry vs 47% Industry, OR 0.29, 95% CI [0.21, 0.41], p < 0.001; S1 Table), in trials registered in DRKS and not on ClinicalTrials.gov (5% DRKS vs 27% ClinicalTrials.gov, OR 0.41,

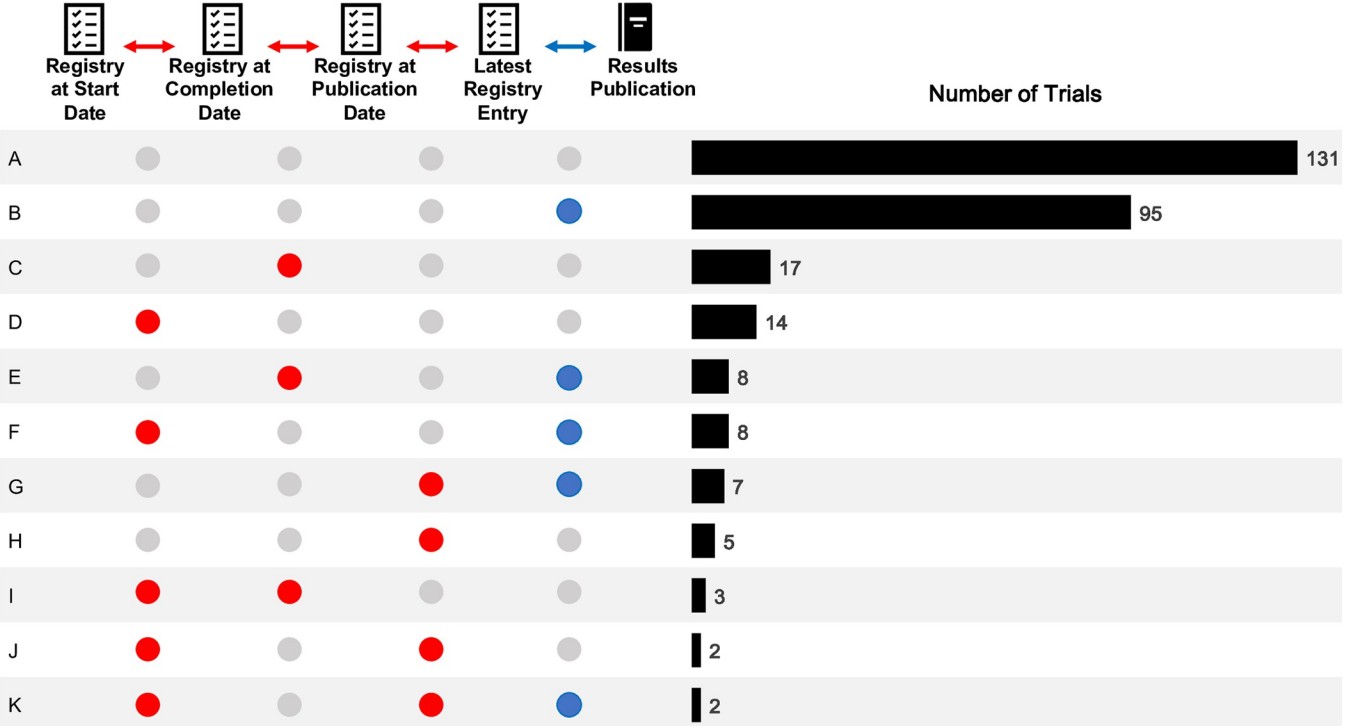

**Fig 3. Number of trials with outcome changes between registry entry versions and/or publications in a random sample of 292 out of 1746 randomized controlled trials.** Registry entry versions were assessed at four subsequent key trial timepoints (within-registry changes; red arrows) and between the latest registry entry and results publication (blue arrow). Rows A to K show each observed combination of outcome change at the different timepoints across the 292 trials. For example, Row A indicates that 131 of 292 trials had no outcome change (45%). Row B indicates that 95 of 292 trials only had a change between the latest registry entry version and the results publication (blue circle; 33%). Changes in rows C, D, H, I and J only occurred within registry entries (red arrow, red circle) while there was agreement of the latest registry entry and the publication (no blue circle), i.e., these changes would not be detected by comparing the results publication to the latest available registry entry (41 of 292 trials, 14%). Row E, F, G and K indicates trials that had a change between the latest entry and the published paper, but also a change at earlier timepoints.

95% CI [0.23, 0.70], p = 0.002), and in trials that were registered later (per registration year; OR 0.74, 95% CI [0.69, 0.80], p < 0.001). No statistically significant association with any other trial characteristic was found.

For registry-publication outcome changes, we identified no statistically significant association with any of the prespecified input variables (S2 Table).

We detected no statistically significant association between within-registry outcome changes and registry-publication outcome changes (S3 Table).

Results were similar in the sensitivity analysis (S4 Table).

## Discussion

Our assessment of all 1746 published clinical trials conducted at German university medical centers over a period of 9 years shows that changes to primary outcomes are common. About one in four trials (23%) have outcome changes within the registry after study start; a major change occurs within one in twelve trials (8%). Of the 292 assessed results publications, 41% have changes between the latest registry entry and the outcomes presented to readers, and 14% of the trials have no easily detectable change, as the changes occurred between different historical versions of the trials' registry entries while the latest entries match the results publications. These changes were minor in 12% and major in 4% of trials. An inspection of only the latest

registry entry version alongside the results publication, without assessment of the entire registration history, would miss a primary outcome change in these trials.

We used a comparatively large sample of German clinical trials registered between 2009 and 2017, with a validated set of accompanying results publications for registry entries, determined by an extensive manual screening process [31, 32]. Our methodology allowed us to detect changes to registered primary outcomes that might have gone unnoticed by other studies, by retrieving all historical versions for trials, and detecting changes using a streamlined workflow. This is different from previous studies, which instead used manual searches [29, 30]. Many trials in our sample had changes to the registry after study completion (9%) or after publication of the first results publication (8%). The changes, for which there may be very plausible reasons, were almost never reported in the results publications. We revealed potential indicators for these practices, with trials that were registered later, trials not registered on ClinicalTrials.gov, and non-industry-sponsored trials having lower odds of within-registry outcome changes. The observation that the frequency of outcome changes declined over time may be explained by recent discussion on clinical trial reporting practices. Regarding the type of registry, the association might be explained by better usability of the ClinicalTrials.gov platform, which might facilitate more frequent changes to registry entries. Regarding sponsorship, our results could indicate better reporting and higher transparency in industry-sponsored trials (which change their registry entries more often than non-industry-sponsored trials). We currently have no sufficient data to quantify the role of selective reporting bias in this sample.

Assessments of outcome changes that rely on a single version of registry entries might underestimate the true prevalence of changes. The ICMJE policy states that any changes to the registration should be explained by authors in a publication [5]. Both the observed lack of reporting on outcome changes in results publications, as well as survey results among manuscript reviewers [6], indicate that reviewers check the registry entry of a clinical trial only in the minority of cases. While registries make change histories available and offer tools to compare different registration versions, and journal editors and reviewers might in some cases use these tools to check the registration histories, the low number of reporting of changes to primary outcomes in the results publications suggests that this is not properly enforced. At the same time, in their current form, historical versions of registry entries are not easily accessible and readable, which makes checking them a time-consuming task. Journals could implement editorial policies that require specialized personnel to assess the historical versions of submitted clinical trials. Clinical trial registries, on the other hand, could implement solutions to identify and mark trials with major changes to their outcomes.

Still, some trials change their outcomes at two or even three different timepoints over the course of the trial. Interestingly, we found not only cases in which primary outcomes gained more detail, but also cases in which primary outcomes were described with less detail than before. Overall, we found 55% of trials to have some form of outcome change over the course of the study. This is somewhat higher compared to other studies, which exhibit a large range for the frequency, with a median of 31% (IQR: 17–45% [8]). This might be due to different methodology, samples, timepoints, or definition of outcome changes. Chan et al. [45], for example, used a more narrow definition of outcome changes (roughly corresponding to our 'major' category). We used a relatively broad definition of primary outcome changes, defining changes to the specificity of primary outcomes as 'minor' changes. Our broader definition and inclusion of more 'minor' outcome changes, some of which might not constitute a large threat to a trial's validity, might have led to our higher estimates.

This study has several limitations. First, we rely on the reporting quality of published studies. In 48 out of 292 publications, the primary outcome was not explicitly described in the results publication, so we had to select the outcome used for sample size calculation or the first

reported outcome instead. A sensitivity analysis excluding these 48 studies supported our main analysis. Second, we only assessed reporting of outcome changes in the main results publications of the respective trials. These changes could also have been published in a protocol or methods paper. We assessed the results publications as they have higher impact, and because current reporting guidelines such as CONSORT state that deviations from the protocol should be reported in the results publication. Third, while we have information about changes that occurred post-completion, we have no data regarding changes in relation to the clinical trial's data lock, that is, the timepoint at which no further information is added to the dataset (which might lie after the completion date). Fourth, the categorization of outcome changes is to some degree subjective, and while the agreement was very high for the within-registry changes ($\kappa$ = 0.81), it was moderate ($\kappa$ = 0.46) for the registration-publication ratings due to the added complexity of identifying the correct outcome as 'primary' and different wording and descriptions between registry and publication. However, conflicting primary outcome definitions and conflicting ratings were resolved in discussions between the reviewers (and, where necessary, a third reviewer), and excluding unclear situations in the above-mentioned sensitivity analysis did not change the main findings. Still, this could become an issue in practice if, for example, different peer-reviewers disagree in their assessment of whether an outcome change has happened or not. Fifth, although we used high quality underlying data reflecting German clinical research output, we only assess a single geographic area, partly reflecting EU/German registration policies. Still, many registration policies are international, and our results point to the possibility that technical details of the registries rather determine whether outcome changes occur. Finally, we only looked at changes between registry entry versions but did not assess how many outcome measures remain underspecified (e.g., no information on the type of measurement, timing, or method of aggregation).

Our analysis also demonstrated the feasibility of an efficient workflow [46] that can be used for future projects and overcome previously described challenges to incorporate historical registry data [47].

To conclude, primary outcome changes are not uncommon in clinical research. Such changes are, in one of four cases, not directly visible upon first inspection of the trial registries and almost never reported in publications. While changes to primary outcomes are not unusual in clinical trials, may be well justified and sometimes desirable, undisclosed changes are a problem. While the problem seems to be ubiquitous, it may be more common in older trials, trials registered in ClinicalTrials.gov, and industry-sponsored trials. Even though trial registries can be an important tool to detect outcome changes, review procedures employed by journals seem to rarely make use of them. Our approach provides a feasible workflow to further investigate this issue. Overall, it is not sufficient to only assess the publication and the latest registry entry–a careful assessment of the full research conduct from conception to publication is needed to ensure trustworthy evidence for clinical decision-making.

## Supporting information

**S1 PRISMA Abstract Checklist. PRISMA 2020 for abstracts checklist.**
(DOCX)

**S1 PRISMA Checklist. PRISMA 2020 checklist.**
(DOCX)

**S1 Fig. Percentage of trials showing primary outcome changes, for each of the within-registry comparisons.** The nodes on the left show changes between the registry entry version at the start date and the registry entry version at the completion date. The nodes in the middle show

changes between the registry entry at the completion date and the registry entry at the publication date. The nodes on the right show changes between the registry entry at the publication date and the latest registry entry. 'No information' means that the registry entries at the two timepoints were the same. This applies to circa 54% of comparisons in the right node, as the registry entry at publication is often the latest entry. The grey bars show the 'flow' of changes: For example, the largest grey bar between the column on the left and the column in the middle indicates that the majority of trials with no changes between the registry entry versions at the start date and completion date also have no changes between the registry entry version at completion date and publication date. For more detailed information, please see our interactive online plot: https://martin-r-h.shinyapps.io/invisibleoutcomechangesfigure/.
(TIF)

**S2 Fig. Number of trials with major outcome changes between registry entry versions and/ or publications in a random sample of 292 out of 1746 randomized controlled trials.** Registry entry versions were assessed at four subsequent key trial timepoints (within-registry changes; red arrows) and between the latest registry entry and results publication (blue arrow). Rows A to H show each observed combination of outcome change at the different timepoints across the 292 trials. For example, Row A indicates that 225 of 292 trials had no major outcome change (77%). Row B indicates that 46 of 292 trials only had a major change between the latest registry entry version and the results publication (blue circle; 16%). Changes in rows C and D only occurred within registry entries (red arrow, red circle) while there was agreement of the latest registry entry and the publication (no blue circle), i.e., these changes would not be detected by comparing the publication to the latest available registry entry (13 of 292 trials, 4%). Rows E, F, G and H indicate trials that had a major change between the latest entry and the published paper, but also a change at earlier timepoints.
(TIF)

**S3 Fig. Number of trials with minor outcome changes between registry entry versions and/ or publications in a random sample of 292 out of 1746 randomized controlled trials.** Registry entry versions were assessed at four subsequent key trial timepoints (within-registry changes; red arrows) and between the latest registry entry and results publication (blue arrow). Rows A to K show each observed combination of minor outcome change at the different timepoints across the 292 trials. For example, Row A indicates that 178 of 292 trials had no minor outcome change (61%). Row B indicates that 61 of 292 trials only had a minor change between the latest registry entry version and the results publication (blue circle; 21%). Changes in rows C, D, E, I and J only occurred within registry entries (red arrow, red circle) while there was agreement of the latest registry entry and the publication (no blue circle), i.e., these changes would not be detected by comparing the publication to the latest available registry entry (39 of 292 trials, 13%). Rows F, G, H and K indicate trials that had a minor change between the latest entry and the published paper, but also a change at earlier timepoints.
(TIF)

**S1 Table. Frequencies, odds ratios (exponentiated regression coefficients) and accompanying p-values for the logistic regression model, with any within-registry outcome change as the output variable (N = 1746 trials).** None of the input variables had missing data, except medical field, where 74 trials could not properly be assigned.
(DOCX)

**S2 Table. Frequencies, odds ratios (exponentiated regression coefficients) and accompanying p-values for the logistic regression model, with any registry-publication outcome change as the output variable (n = 292 trials).** None of the input variables had missing data,

except medical field, where 9 trials could not properly be assigned.
(DOCX)

**S3 Table. Frequencies, odds ratios (exponentiated regression coefficients) and accompanying p-values for the logistic regression model, with any within-registry outcome change as the output variable (n = 292 trials) and any within-registry outcome change as input variable.** The input variable did not have missing data.
(DOCX)

**S4 Table. Primary outcome changes in 1746 randomized controlled trials completed between 2009 and 2017 in German university medical centers.** This table includes a sensitivity analysis for the registry-publication changes, including only publications with explicitly named primary outcomes (n = 243).
(DOCX)

## Author Contributions

**Conceptualization:** Martin Holst, Daniel Strech, Lars G. Hemkens, Benjamin G. Carlisle.

**Data curation:** Martin Holst.

**Formal analysis:** Martin Holst, Martin Haslberger.

**Funding acquisition:** Daniel Strech.

**Investigation:** Martin Holst, Martin Haslberger, Samruddhi Yerunkar.

**Methodology:** Martin Holst, Martin Haslberger, Lars G. Hemkens, Benjamin G. Carlisle.

**Software:** Benjamin G. Carlisle.

**Supervision:** Lars G. Hemkens, Benjamin G. Carlisle.

**Validation:** Benjamin G. Carlisle.

**Visualization:** Martin Holst, Martin Haslberger.

**Writing – original draft:** Martin Holst, Martin Haslberger.

**Writing – review & editing:** Samruddhi Yerunkar, Daniel Strech, Lars G. Hemkens, Benjamin G. Carlisle.

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
