## [Editor Report · Decision Letter 0]

2 Mar 2023

Dear Dr Holst, 

Thank you for submitting your manuscript entitled "Hidden changes to prespecified primary outcomes of clinical trials completed between 2009 and 2017 in German University Medical Centres: A meta-research study" for consideration by PLOS Medicine.

Your manuscript has now been evaluated by the PLOS Medicine editorial staff and I am writing to let you know that we would like to send your submission out for external peer review.

Please re-submit your manuscript within two working days, i.e. by Mar 06 2023 11:59PM.

Kind regards,

Pippa

Philippa Dodd, MBBS MRCP PhD

PLOS Medicine

---

## [Decision Letter · Decision Letter 1]

3 Jul 2023

Dear Dr. Holst,

Thank you very much for submitting your manuscript "Hidden changes to prespecified primary outcomes of clinical trials completed between 2009 and 2017 in German University Medical Centres: A meta-research study" (PMEDICINE-D-23-00549R1) for consideration at PLOS Medicine. 

[LINK]

In light of these reviews, I am afraid that we will not be able to accept the manuscript for publication in the journal in its current form, but we would like to consider a revised version that addresses the reviewers' and editors' comments. Obviously we cannot make any decision about publication until we have seen the revised manuscript and your response, and we plan to seek re-review by one or more of the reviewers. 

We expect to receive your revised manuscript by Jul 24 2023 11:59PM. Please email us (plosmedicine@plos.org) if you have any questions or concerns.

We look forward to receiving your revised manuscript. 

Sincerely,

Philippa Dodd, MBBS MRCP PhD

PLOS Medicine

plosmedicine.org

GENERAL

Please respond to all editor and reviewer comments detailed below in full.

Please justify the inclusion of studies published only from Germany. 

As you report an analysis of existing studies as metadata (also noting the use of ‘Numbat Systematic Review Manager’ for data extraction), we suggest that you consider reporting your study according to PRISMA for SRMAs instead of STROBE. The PRISMA guidelines can be found at the EQUATOR site, http://www.equator-network.org/reporting-guidelines/prisma/ We acknowledge that some aspects of PRISMA reporting may not apply here.

Please provide the completed PRISMA checklist. 

When completing the checklist, please use section and paragraph numbers, rather than page/line numbers as these often change in the event of publication.

Please add the following statement, or similar, to the Methods: "This study is reported as per the Preferred Reporting Items for Systematic Reviews and Meta-Analyses (PRISMA) guideline (S1 Checklist)."

Please temper the language used throughout, including the use of the word ‘hidden’ in the title, and ensure only objective reporting of your findings and their implications.

Comments from the Academic Editor - it would be important for the authors to avoid words like "Hidden" in the title, which implies malfeasance when we don't know either way about that. A chance for major revision seems reasonable to me.

DECLARATIONS

Lines 44-60, please remove these statements from the main manuscript and include only in the manuscript submission form. They will be compiled as metadata in the event of publication.

TITLE

Please remove the word ‘hidden’ from your title and revise it according to PLOS Medicine's style. Your title must be nondeclarative and not a question. It should begin with main concept if possible. "Effect of" should be used only if causality can be inferred, i.e., for an RCT. Please place the study design ("A randomized controlled trial," "A retrospective study," "A modelling study," etc.) in the subtitle (ie, after a colon).

ABSTRACT

Please report your abstract according to PRISMA for abstracts, following the PLOS Medicine abstract structure (Background, Methods and Findings, Conclusions). Further details can be found via this link http://www.plosmedicine.org/article/info:doi/10.1371/journal.pmed.1001419

Abstract Background: 

Please provide the context of why the study is important. The final sentence should clearly state the study question.

Abstract Methods and Findings:

Line 68 – please amend to read ‘randomized controlled trials’

Please ensure that all numbers presented in the abstract are present and identical to numbers presented in the main manuscript text.

Please provide the eligibility criteria, and synthesis/appraisal methods. 

Please justify the choice of start and end dates of the studies included/searched for. 

Please quantify the main results p values as well as with 95% CIs.

When reporting p values please report as p<0.001 and where higher as p=0.002, for example.

Suggest reporting statistical information as follows – ‘OR 0.29; 95% CI [0.21, 0.41], p<0.001’

Please define OR at first use for the reader.

Were any variable factors adjusted for in your analyses? If so, please detail which.

In the last sentence of the Abstract Methods and Findings section, please describe the main limitation(s) of the study's methodology.

Abstract conclusions

Please temper the language used here and address the study implications without overreaching what can be concluded from the data; the phrase "In this study, we observed ..." may be useful.

Please interpret the study based on the results presented in the abstract, emphasizing what is new without overstating your conclusions.

Please avoid vague statements such as "these results have major implications for policy/clinical care". Mention only specific implications substantiated by the results.

Please avoid assertions of primacy ("We report for the first time....")

AUTHOR SUMMARY

At this stage, we ask that you include a short, non-technical Author Summary of your research to make findings accessible to a wide audience that includes both scientists and non-scientists. The authors summary should consist of 2-3 succinct bullet points under each of the following headings:

• Why Was This Study Done? Authors should reflect on what was known about the topic before the research was published and why the research was needed.

• What Did the Researchers Do and Find? Authors should briefly describe the study design that was used and the study’s major findings. Do include the headline numbers from the study, such as the sample size and key findings. 

• What Do These Findings Mean? Authors should reflect on the new knowledge generated by the research and the implications for practice, research, policy, or public health. Authors should also consider how the interpretation of the study’s findings may be affected by the study limitations. In the final bullet point of ‘What Do These Findings Mean?’, please describe the main limitations of the study in non-technical language.

The author Summary should immediately follow the Abstract in your revised manuscript. This text is subject to editorial change and should be distinct from the scientific abstract. Please see our author guidelines for more information: https://journals.plos.org/plosmedicine/s/revising-your-manuscript#loc-author-summary

INTRODUCTION

Please address past research and explain the need for and potential importance of your study. Indicate whether your study is novel and how you determined that. If there has been a systematic review of the evidence related to your study (or you have conducted one), please refer to and reference that review and indicate whether it supports the need for your study.

Please remove the phrase ‘cherry-picking’

Line 116 – please refrain from using the word ‘hidden’. 

Lines 130 onwards – please replace square brackets with circular (in-text reference callouts should be placed in square brackets – see below under REFERENCES)

Has there been a systematic review of the evidence related to your study (or if you have conducted one)? If so, please refer to and reference that review and indicate whether it supports the need for your study.

METHODS and RESULTS

Line 142 – please place citation in square brackets

Line 143 – ‘interventional studies’ please replace with ‘randomized controlled trials’ 

As above, please justify the inclusion of data from these specified time points

As above, please justify the inclusion of only studies originating from Germany

Line 167, 213 – please use an alternative word for ‘downloaded’, ‘extracted’ perhaps? Please check and amend throughout.

Line 170 onwards – please reserve the use of square brackets for in-test reference callouts

Line 213 onwards – details of the software packages would be better placed earlier at line 165 where you describe data extraction and processing.

Line 216 – suggest reporting according to PRISMA instead of STROBE. 

DISCUSSION

Please remove the sub-heading ‘conclusion’ form the end of the discussion.

As above, please refrain from using the word ‘hidden’.

REFERENCES

For in text reference callouts please place citations in square brackets.

Please list up to but no more than 6 author names followed by et al.

Please ensure that all web references include an access date.

Please see our website for other reference guidelines https://journals.plos.org/plosmedicine/s/submission-guidelines#loc-references

TABLES

Please ensure that all tables area affiliated to an appropriate caption which clearly describes the table contents without the need to refer to the text. Please ensure all abbreviations are defined for the reader.

Suggest presenting the trial characteristics as table 1 followed by the definitions of discrepancies as table 2

Table 1 – outcome discrepancies – these seem rather vaguely defined, suggest additional nuance to the categorization. When reporting minor discrepancies the word ‘significant’ is used implying a major discrepancy and is also not very quantified. Please revise this table including more nuanced detail of the categorization applied.

Table 2 – The category ‘other’ constitutes the second large group of studies in your dataset. Could other be sub-categorized for additional information?

Table 3 – for the main outcomes measures please indicate whether your analyses are adjusted or unadjusted and if adjusted please present the unadjusted analyses for comparison. Please report p as p<0.001 and where higher the exact p value. Please define the meaning of the lettering in bold type-face

FIGURES

Please ensure that all figures are associated with a caption that clearly describes the figure content without the need to refer to the manuscript text. Please ensure all abbreviations and the meaning of dots, lines and bars are clearly defined for the reader.

Figure 1 – if the end point in this flow diagram is ‘journal publication’ does the earlier ‘publication’ refer to the protocol publication? Please clarify/revise for the reader.

Figure 3 is striking but rather confusing. It details percentages but does not give a scale to determine the percentage. What do the different widths of shaded grey areas and lines represent? Perhaps a key for the colored bars could be provided as a legend for clarity. 

Figure 4 – please provide the unit of measurement for the ‘discrepancies across timepoints’ total number? The label of the y axis could be closer to the graph, please revise.

SUPPLEMENTARY FIGURES

Please ensure that all figures are associated with a caption that clearly describes the figure content without the need to refer to the manuscript text. 

Please ensure all abbreviations and the meaning of dots, lines and bars are clearly defined for the reader. 

Please ensure axis labels are placed appropriately alongside the relevant axis.

 Please ensure that all axes have a defined unit of measurement e.g., total number. 

If using ‘disc.’ as an abbreviation please ensure that it is defined,

SUPPLEMENTARY TABLES

Where p values are presented, please report as p<0.001 and where higher as the exact p value. Please also include 95% CIs where you report p values.

Comments from the reviewers:

Reviewer #1: Statistical review

This paper reports an investigation of discrepancies between registration entries and publications in clinical trials in terms of primary outcomes. The authors demonstrate there are a considerable number of discrepancies and that the type of study is associated with the chance of discrepancy. The statistical methods used are (appropriately) straightforward. I had some comments on the reporting below:

1. Abstract, '[1] Primary outcome discrepancies between registry entries at key study milestones and [2] the first results publication." - possibly moving the [1] might help this be a bit clearer, e.g.: 'Primary outcome discrepancies between registry entries [1] at key study milestones and [2] the first results publication."

2. Abstract: "Only 1% of discrepancies were reported in the publications (2/161, 95% CI [0%, 4%])." - can it be made clearer from the abstract why 161 is the denominator here?

3. Abstract (and results): the ORs are <1 indicating registry discrepancy is less likely, not more likely. 

4. Methods (and discussion): the agreement for registry-publication results seems fairly low - this is presumably prior to discussion to resolve disagreements? This does seem a limitation to applying these results in practice for identifying whether there were inconsistencies when reviewing future publications: some reviewers may identify an issue and some may not. Perhaps this could be mentioned in the discussion. 

5. Discussion: I presume the authors did not extract information on whether results were significant with the discrepant outcomes. This would be interesting information if it exists (e.g. trials with discrepant outcomes were more likely to report significant results) as it would imply that this might involve cherry-picking.

6. Supplementary table 2 - it would be useful to provide the 95% CI for the odds ratios in this table.

James Wason

Reviewer #2: This is an interesting meta-research study investigating discrepancies between clinical trial protocols in registries vs published reports. Overall, the findings are likely to be of interest to trialists and other readers, though the manuscript and reporting could be improved in a number of areas. Please see comments below.

# General comments

The scope of this study is substantial, with a large number of outcomes, statistical tests, and presentation in figures that are not common in most fields of research. In addition, the writing uses a number of assumed terms and knowledge, which may be problematic for the broad readership of this journal. And the writing is hard to follow in places. 

* Suggest to edit the manuscript to simplify wording and tighten the writing, remove/minimise jargon, and clearly define terms at first use. 

* Consider displaying data using figures that are more recognisable in medical research. If not possible, substantially expand the figure legends to orientate figures to readers who may not be research-active in this area.

# Specific comments

## Abstract

L64. 'primary outcome discrepancies' can read as jargon: consider wording as e.g. 'differences between primary outcomes reported in trial registries and in published reports (i.e. primary outcome discrepancies)'

L64. The first sentence also seems odd: It is worded as "to asses.. discrepancies within registry records.." but does not state what registry records are discrepant to. Please refine.

L77. We considered discrepancies [ADD: to be] major if primary outcomes were [DEL: newly] added, dropped, [EDITED: replaced by secondary outcomes], or [EDITED: used to replace secondary outcomes].

L82 and in Tables, Results throughout manuscript: A 95% CI represents a range of values over which the estimate falls. Please format all confidence intervals as e.g. [35% to 47%], instead of using commas to separate the upper and lower limits. This makes clear to readers that the confidence interval is a range of values. 

## Introduction

L98. The word 'critical' is mentioned 3 times in the first 6 lines, which seems an overuse of the word and dilutes its seriousness. Please rephrase to keep the emphasis on what is important and de-emphasise less important points. 

L103. .. asking [ADD: authors] to 

L106: 'changed': what does the change represent? I.e. changed from __ to __? 

L107: What are the implications of multiple testing? 

For that matter, on my count there are 7 + 28 + 27 + 1 + 7 = 70 tests in this study, in the main paper and supplementary tables. This large number of tests highlights the exploratory nature of this study, and also makes it prone to limitations of multiple testing. 

Why might this study fair better than other studies with substantial multiple tests?

L115: What is being considered?

L122-124. The flow between sections of this sentence is not clear

'most studies published to date did not XXX, assessed XXX, or the latest entry ...' does not make sense. Please fix. 

L126-129: It seems unclear what value these statements add to this paragraph, that there is variability between registered protocols and trial reports. Consider dropping them, or revise to connect the ideas more strongly to the sentences above.

L130-137. The large number of aims and lengthy articulation seems to dilute the strength of the key message of this study, and make the paper look overambitious. 

Consider if the aims can be simplified to focus the reader on key aspects of this study. 

## Methods

L152, 158. Why would a study be regarded as 'completed' if the trial has an 'Unknown status'? How can investigators be sure it was actually completed? 

L185. Please explain clearly 'If the reviewers could not find the correct publication, they excluded the publication'

L205. 'proportions': should be percentages?

L209. Why are these variables regarded as 'predictors'? Are future events being predicted from preceeding knowledge on study phase, sponsor, ... intervention?

## Results

L230. Consider expanding on 'changed to or from a secondary outcome'

L230. Percentages reported as integer values in Results text, but to 2 dec place in Tables. Please make consistent: report percentages in Tables as integers. It is unlikely that precision to 2 dec place of a percent is meaningful. 

L259. I may have missed it, but the positive result for trials registered on ClincialTrials.gov seems an unusual finding and does not seem to be sufficiently explained. Is this finding because most trials tend to be registered on this registry anyway?

## Discussion

Show paragaraph separation throughout, in the Discussion.

L272. should be '[have] discrepancies'

L288. Replace 'likelihood' with 'odds'. 'Likelihood' has a special meaning in statistics. 

L298. Reviewers already review on volunteer basis under time-poor conditions. What incentives are there for reviewers to do more work, as suggested here?

L310. 'defining added or omitted specificity': rephrase to make clear

L323. Delete 'that'

L325. Rephrase 'did not only reveal these research findings'; clunky. 

## References

Please check reference formatting is consistent -- there are some inconsistencies throughout

* Make all titles sentence case (e.g. ref 3)

* Remove month

* Check all abbreviated journal names are correct (e.g. ref 26)

* Fix formatting ref 37

## Tables

Table 1.

* 'Multiple categories' of what? What 'timepoints'?

* State 'outcome' if it refers to an outcome, not just 'primary' or 'secondary'

* 'aggregation' of what?

* 'significant parts' of what?

* What is the meaning of the curly brackets?

* Define RECIST

Table 2

* Make separate headers for median (IQR) and N (%). Break the table afdter first 3 rows to add the N (%) header

* decrease width of 2nd column so row items in 1st column do not break across lines

* State 'Other or not provided' as 'Other or Not provided' to be consistent

* Drop the last row. The total score is the sum of percentages within rows, not percentages across all rows. 

Table 3 and Supplementary Table 4. 

* Report percentages as integers (i.e. no dec pla) to be consistent with manuscript

* Make formatting consistent: capitalise 'Recruitment', 'Post-completion', 'Post-publication'

* Define why font is bold

* Define statistics in last column

Supplementary Tables 1, 2

* Define statistics

* Report percentages as integers

* Make column labels consistent with capitalisation

* Why does 'p-Value' have a capital V?

* Define 'Intercept'

* Report p values less than 0.001 as 'p<0.001'; there's no need to show p values to 16 decimal places

Please pay more attention-to-detail in Tables and Figures. 

## Figures

Overall, I found the Figures the least clear and the hardest section of this manuscript to follow, which is a shame since so much information is conveyed in figures. Please address the comments.

Fig 1. 

* Capitalise t in 'time'. Make 'Publication' single line

* In Legend, define vertical dotted lines, vertical dashed line, boundaries of blue box. Explain meaning of bidirectional red arows; why do arrows need to be red? Why does 'Journal publication' need an icon, but the other text does not?

* Expand the legend substantially to talk readers through the figure and orientate them. Each figure needs to 'stand-alone'.

Note, any qualitative difference in patterns in figures can be interpreted to have a meaning, otherwise no difference is needed. So if differences are shown but there is no meaning, it leaves the impression that the figure is sloppy and rushed, not clear and informative. 

Fig 2. 

* Why are lower 3 boxes in grey but the others in black? Make consistent

* In Exluded trials: 

 * rephrase 'publication' with 'published'

 * why were time limits of 2 and 5 years chosen? Was this explained in the text? 

 * by definition, a clinical trial that is not randomised or does not have an intervention can't be a 'trial'. Does this suggest that observational studies are registered on ClinicalTrials.org? Would it be more accurate to use 'study' here?

 * 'samples' should be 'registries'?

* Also, expand the Legend to provide details on information in boxes. 

Fig 3. Very hard to follow. At present, the figure is so unintuitive and uninformative that it seems better to drop it than keep it in. 

* Does the height of the bar have meaning? Are bars drawn to scale? 

* Label the y axis, provide axis and axis markers. 

* Join the bars so they are continuous

* Define 'Major' and 'Minor'. Define 'Start' vs 'Completion' I.e. start of what? 

* Define 'Pub'

In legend,

* Explain, what is this figure meant to show? What are the bars supposed to represent, and what are the curved grey regions defined as and meant to show.

* Expand the legend substantially to explain this figure to readers. 

Fig 4, and similar Supplementary Figs.

* This seems to be a specialised plot for a niche area, but will be new to general readership of this journal.

Add an introduction to the legned to explain how this figure is to be read and interpreted, what the vertical bar heights indicate, what the horizontal bar heights indicate, and what the dots and connecting lines indicate. Provide an example reading of the matrix. 

* At the moment, I cannot match the numbers reported in the legend to bar heights etc. in the figure: e.g. where did 41 (14%) come from?

Reviewer #3: To Authors

I found this difficult to follow, I'm afraid. On this reading, I was left suspecting that readers will think: "Is this a German problem?", "Is this an old problem?" and/or "Is this a real problem?" I didn't fully understand what had been done nor, often, why. The comments are all intended constructively and I hope they come across that way. Below are listed detailed comments.

===CRITICAL===

1. 

:: Section :: General

:: Comment :: I'm afraid I didn't get the importance of this work. What am I missing? Is the suggestion that people don't change outcome measures if they need to or that they report better or that registers or journals should take some action?

2. 

:: Section :: General

:: Comment :: I thought the terminology felt harsh and misleading, particularly "discrepancy" and "hidden". It's only discrepant if the paper differs to the most recent registry entry. It's only hidden if the change doesn't appear in the registry. (I spend most of my career researching how to improve the way do clinical trials - there's lots of scope for improvement - so I was surprised by how the wording of this manuscript provoked a defensive reaction in me.) This requires a big re-think, including in Table 1.

===MODERATE===

3. 

:: Section :: Background

:: Comment :: Why is it a problem to look through the history of changes? This is the point of history of changes?

4. 

:: Section :: Throughout

:: Comment :: I always understand that an "outcome" is how the patient does and an "outcome measure" is how you measure the outcome. An endpoint would be the same as an outcome measure. The use of outcome (as a shorthand for "outcome measure"?) here seemed confusing.

5. 

:: Section :: Methods

:: Text ref :: "We based our analyses on two published datasets (from the IntoValue projects, 31,32) containing all interventional studies completed at German University Medical Centres between 2009 and 2017."

:: Comment :: Chosen for convenience rather than relevance?

6. 

:: Section :: Methods

:: Text ref :: "The trials had been registered in either ClinicalTrials.gov or the Deutsches Register Klinischer Studien (DRKS), which is the WHO primary trial registry for Germany."

:: Comment :: How many were registered in both? I suspect quite a few. I would have liked the results broken down by registry: some have better supported updating the entries than others.

7. 

:: Section :: Methods

:: Text ref :: " We included any study that: 1) has a registry entry in either the ClinicalTrials.gov or the DRKS database, 2) was completed between 2009 and 2017 according to the trial status described as 'Completed', 'Unknown status', 'Terminated', or 'Suspended' (ClinicalTrials.gov), or 'Recruiting complete, follow-up complete', 'Recruiting stopped after recruiting started', or 'Recruiting suspended on temporary hold' (DRKS), 3) reported in the registry that a German University Medical Centre was involved (i.e., mentioned as responsible party, lead/primary sponsor, principal investigator, study chair, study director, facility, collaborator, or recruitment location; for definitions see 31,32), 4) has published results, i.e., a full-text publication of the trial results was found by the search methods described in the IntoValue protocols (31,32), 5) is registered as a randomised trial."

:: Comment :: First, are these joined together with "and" statements or "or" statements? I suspect the form but it's not clear. Second, the selection criteria confuse "completing accrual" with "completing the trial". Which is actually meant? 

8. 

:: Section :: Methods

:: Text ref :: "For feasibility, we drew a random sample of 300 publications, from which three reviewers (MRH, MH, SY) extracted primary outcomes."

:: Comment :: I accept couldn't do all of them but why 300?

9. 

:: Section :: Methods

:: Text ref :: "If the reviewers could not find the correct publication, they excluded the publication…"

:: Comment :: If it's not the right publication, how could discrepancy be judged?

10. 

:: Section :: Results

:: Text ref :: "Of 1746 trials, 393 (23%) had an outcome discrepancy reported within the registry (Table 3; Figure 3), 142 trials (8%) had major discrepancies, i.e., a primary outcome was newly added, dropped, or changed to or from a secondary outcome."

:: Comment :: When? This is really important for the documented-history vs unreported-discepancy.

11. 

:: Section :: Methods

:: Comment :: Why look only at the results paper? Many trials will publish a methods paper too. The choices may be better documented in these papers (where there is space) than in results papers (where journals, particularly the well-read journals, are very restrictive in space).

12. 

:: Section :: Results

:: Text ref :: "Out of 292 randomly selected trials among the 1746 trials, 120 (41%; 95% CI [35%, 47%]) had a discrepancy between the latest registry entry and the publication, of which 54 (18%; 95% CI [14%, 237 23%]) were major and 75 (26%; 95% CI [21%, 31%]) were minor discrepancies."

:: Comment :: The 120 / 292 is the interesting part of this manuscript. This should be the focus. The rest reads a bit like noise to me. Broken by registry would be good.

13. 

:: Section :: Discussion

:: Text ref :: "The ICMJE policy states that any changes to the registration should be explained by authors in the publication (4)."

:: Comment :: Why "the publication" rather than "as part of the publications"? ICJME aren't always right, of course. They don't enforce CONSORT, they are too strict on authors and they got data sharing wrong, so I suspect "a" rather than "the" should be fine.

14. 

:: Section :: Discussion

:: Text ref :: "Still, some trials change their outcomes at two or even three different timepoints over the course of the trial."

:: Comment :: A reminder of examples here would be good. I've been involved recently with some very long-term trials. During the course of the trials, external projects showed that earlier outcome measures could be used as a surrogate for the existing primary outcome measures so the trial team (without reference to accumulating comparative data, after considering the implications for sample size and timelines and after peer review) formally amended the outcome measures. This saved a couple of years and helped patients sooner. The rational for the changes was documented in methodology papers and these were referenced in the main results papers. These trials were registered both in clinicaltrials.gov which allows changes to be captured and ISRCTN.com which doesn't allow updates (or didn't at the time). How would these examples be judged here.

15. 

:: Section :: Discussion

:: Text ref :: "Interestingly, we found not only cases in which primary outcomes gained more detail, but also cases in which primary outcomes were described with less detail than before."

:: Comment :: I don't really know what this means.

16. 

:: Section :: Figure 1

:: Comment :: I didn't really understand this. Why does the blue box extend beyond the dotted line?

17. 

:: Section :: Figure 2

:: Comment :: This is not well presented. Sometimes a splitting line means the patients go down just one of the paths and other times it means the patients go down both.

18. 

:: Section :: Figure 3

:: Comment :: No scale, no definitions, no timelines. I completely don't understand this.

19. 

:: Section :: Table 2

:: Comment :: I would have liked a column for all trials (as shown) and an extra one for the 300 samples and another for the ones selected as odd.

20. 

:: Section :: Table 3

:: Comment :: What is a post-publication discrepancy? What differs from what in those instances? Is this a change in the registry after a first results paper in anticipation of another longer-term paper?

21. 

:: Section :: Supp Tables

:: Comment :: The lack of totals make these difficult to use. What are these p-values? And they are the separate by row rather than looking for patters without predictors?

[LINK]

---

## [Decision Letter · Decision Letter 2]

21 Sep 2023

Dear Dr. Holst,

Thank you very much for re-submitting your manuscript "Frequency of ‘invisible’ changes to prespecified primary outcomes of clinical trials completed between 2009 and 2017 in German university medical centers: A meta-research study" (PMEDICINE-D-23-00549R2) for review by PLOS Medicine.

I have discussed the paper with my colleagues and the academic editor and it was also seen again by 2 reviewers. I am pleased to say that provided the remaining editorial and production issues are dealt with we are planning to accept the paper for publication in the journal.

[LINK]

We look forward to receiving the revised manuscript by Sep 28 2023 11:59PM.   

Sincerely,

Philippa Dodd, MBBS MRCP PhD

PLOS Medicine

plosmedicine.org

Requests from Editors:

GENERAL

Thank you for your detailed responses to previous editor and reviewer comments. Please see below for further comments that we require you address in full prior to publication.

* We require that you refrain from any reference to ‘hidden’ or ‘invisible’ changes which could mislead the reader and implies an intentional lack of transparency. Indeed, you were able to find these changes in available documents. We suggest ‘late changes’ or ‘subsequent amendments’ or similar, as an alternative. In addition, the tone throughout is rather accusatory – there are many reasons why amendments may be necessary to trial protocols – a more objective and balanced presentation of only the facts supported by the data you present are required. This is a prerequisite to publication. *

** In quite a few studies, primary outcomes were modified AFTER recruitment began and, very few papers transparently reported protocol changes per ICMJE. Please clarify whether you checked when primary (or other) outcome changes occurred relative to data lock. This is a key point of the manuscript which we agree warrants specific discussion. Please include. **

TITLE

Please revise your title to refrain from using the word ‘invisible’, we suggest, ‘Frequency of changes to prespecified primary outcomes of clinical trials completed between 2009 and 2017 in German university medical centers: A meta-research study’.

COMPETING INTERESTS

All authors must declare their relevant competing interests per the PLOS policy, which can be seen here:

https://journals.plos.org/plosmedicine/s/competing-interests

For authors with ties to industry, please indicate whether any of the interests has a financial stake in the results of the current study.

ABSTRACT

Line 69 – in reference to ‘major’ changes – please briefly explain what is considered to be ‘major’ in this study. 

Line 76/77 – ‘often major’ could be considered an overstatement in light of the data presented. Please interpret the study based on the results presented in the abstract, emphasizing what is new without overstating your conclusions.

AUTHOR SUMMARY

Line 87 – suggest ‘allow assessment of…’

Line 90 – suggest ‘unapparent’ instead of invisible.

Line 91-92 – suggest ‘Our objective was to investigate how often primary outcomes are changed in the trial registry over the course of a trial and, what proportion of the changes were unapparent.’

Line 100 – please revise the use of the terms ‘visible’ and ‘invisible’.

Line 102 – suggest ‘Only 1% of changes (2 trials), were reported in the corresponding publications.

Line 106 – as above how do you define ‘major’? It would be helpful to include (brief) details under the ‘what did the researchers do and find’ section.

Line 107 – please remove the word ‘invisible’.

INTRODUCTION

Throughout, please remove reference to ‘invisible’.

Line 113 – suggest ‘validity’ or similar, in place of ‘trustworthiness’.

METHODS and RESULTS

Throughout please remove all reference to ‘invisible’.

Line 202 onwards – you only briefly refer to the classification system you use to determine ‘severity’ of the change identified but, you give no details other than referring the reader to the table. It is a little unfair to expect the reader to look elsewhere for this. Please provide brief details in the text.

DISCUSSION

Throughout please remove all reference to the term ‘invisible’.

The editorial team agree that you should present a more balanced and nuanced discussion of the drivers of protocol/primary outcome changes, the very vast majority of which are legitimate. 

It would also be helpful to contextualize the discussion in context of the ‘severity’ scoring system you apply to the changes that you identify.

Line 370 – would ‘not uncommon’ be a better term?

Line 371 – ‘often’ is somewhat subjective and perhaps an overstatement. Suggest simply stating ‘one in four’.

REFERENCES

Please ensure that all web references include an ‘Accessed [date]’ instead of, ‘Cited [date]’. Please check and amend throughout.

Please remove all instances of ‘[internet]’.

SUPPORTING INFORMATION

Table S1 - please detail the meaning of the asterisk against the p values in an appropriate footnote.

Table S4 – please revise ‘No’ to ‘N’ (or use the full word ‘number’) and define in the footnote.

PRISMA Checklist – thank you for including the checklist. Please revise the final column to detail section and paragraph numbers as opposed to page numbers, as these often change at publication.

SOCIAL MEDIA

To help us extend the reach of your research, please detail any Twitter handles you wish to be included when we tweet this paper (including your own, your coauthors’, your institution, funder, or lab) in the manuscript submission form when you re-submit the manuscript.

Comments from Reviewers:

Reviewer #1: Thank you to the authors for addressing my previous comments well. I have no further issues to raise.

Reviewer #2: The authors have made substantial changes to the manuscript. A few smaller edits for clarity remain.

# Specific comments

## Abstract

L46. Rephrase as: Clinical trial registries [edit: allow to assess -> make transparent] deviations...

L56-60. Rephrase so sentence stem reads consistently with all enumeration points 

i.e. We determined changes of primary outcomes ... and (3) [edit: 'invisible' changes ... -> that occurred after trial start with no change between last registry entry version and publication (i.e. invisible changes) ...]

## Author summary

L86. Rephrase: They [edit: allow to assess -> make transparent] how closely...

L132-134. Rephrase sentence: [edit: Only a closer look ... not easily accessible -> These changes would only be discovered on closely investigating the histroy of changes in trial registries.]

## References

Please check all refs are formatted consistently. 

Numerous refs need the article title changed to sentence case e.g. Refs 3, 4, 6, 9, 11, 12, ....

[LINK]

---

## [Editor Report · Decision Letter 3]

3 Oct 2023

Dear Dr Holst, 

On behalf of my colleagues and the Academic Editor, Professor Aaron Kesselheim, I am pleased to inform you that we have agreed to publish your manuscript "Frequency of multiple changes to prespecified primary outcomes of clinical trials completed between 2009 and 2017 in German university medical centers: A meta-research study" (PMEDICINE-D-23-00549R3) in PLOS Medicine.

Prior to publication we require that you address the following:

1) Thank you for revising the title. We suggest removing the word ‘frequency’ as this is partially declarative. Considering the comments in your previous response we suggest referring to the registry histories in the title, ‘Historic registry changes to prespecified primary outcomes of clinical trials completed between 2009 and 2017 in German university medical centers: A meta-research study.’ Or similar.

2) Line 44 - thank you for confirming that no competing interests exist. Please remove the competing interest statement from the main manuscript and include only in the submission form. 

3) Line 110 – suggest ‘…published results…’

PRESS

Thank you again for submitting to PLOS Medicine. It has been a pleasure handling your manuscript, we look forward to publishing your paper. 

Best wishes,

Pippa 

Philippa Dodd, MBBS MRCP PhD  

PLOS Medicine

pdodd@plos.org

plosmedicine@plos.org